# CLIP4HOI: Towards Adapting CLIP for Practical Zero-Shot HOI Detection

**Yunyao Mao**[1]    **Jiajun Deng**[2]    **Wengang Zhou**[1†]    **Li Li**[1]    **Yao Fang**[3]    **Houqiang Li**[1†]

[1]CAS Key Laboratory of Technology in GIPAS, EEIS Department,
University of Science and Technology of China
[2]The University of Adelaide, AIML    [3]Merchants Union Consumer Finance Company Limited
`myy2016@mail.ustc.edu.cn, jiajun.deng@adelaide.edu.au, zhwg@ustc.edu.cn`
`lil1@ustc.edu.cn, fangyao@mucfc.com, lihq@ustc.edu.cn`

## Abstract

Zero-shot Human-Object Interaction (HOI) detection aims to identify both seen and unseen HOI categories. A strong zero-shot HOI detector is supposed to be not only capable of discriminating novel interactions but also robust to positional distribution discrepancy between seen and unseen categories when locating human-object pairs. However, top-performing zero-shot HOI detectors rely on seen and predefined unseen categories to distill knowledge from CLIP and jointly locate human-object pairs without considering the potential positional distribution discrepancy, leading to impaired transferability. In this paper, we introduce CLIP4HOI, a novel framework for zero-shot HOI detection. CLIP4HOI is developed on the vision-language model CLIP and ameliorates the above issues in the following two aspects. First, to avoid the model from overfitting to the joint positional distribution of seen human-object pairs, we seek to tackle the problem of zero-shot HOI detection in a disentangled two-stage paradigm. To be specific, humans and objects are independently identified and all feasible human-object pairs are processed by Human-Object interactor for pairwise proposal generation. Second, to facilitate better transferability, the CLIP model is elaborately adapted into a fine-grained HOI classifier for proposal discrimination, avoiding data-sensitive knowledge distillation. Finally, experiments on prevalent benchmarks show that our CLIP4HOI outperforms previous approaches on both rare and unseen categories, and sets a series of state-of-the-art records under a variety of zero-shot settings.

## 1   Introduction

Human-Object Interaction (HOI) detection aims at locating paired humans and objects and identifying their interactions. It finds applications in several downstream tasks, including surveillance, robotics, and human-computer interaction. Despite significant progress in recent years, in most previous works, the task of HOI detection has been conventionally restricted to predefined HOI categories. Given the flexibility of verb-object combinations in human-object interaction scenarios, obtaining a comprehensive dataset is laborious and time-consuming. Moreover, the obtained detectors also struggle to generalize to unseen HOI categories, even if both verb and object in the novel category are present (but never composed together) in the training set. These facts motivate and highlight the exploration of zero-shot HOI detection, which targets to development of transferable HOI detectors.

Latest zero-shot HOI detectors [31, 44, 46, 48] seek to harness the general visual and linguistic knowledge of CLIP [38] for identifying novel HOIs. Despite notable advancements over conventional compositional learning approaches [2, 15, 17–19, 34, 36], these methods exhibit certain limitations.

---

[†]Corresponding authors: Wengang Zhou and Houqiang Li

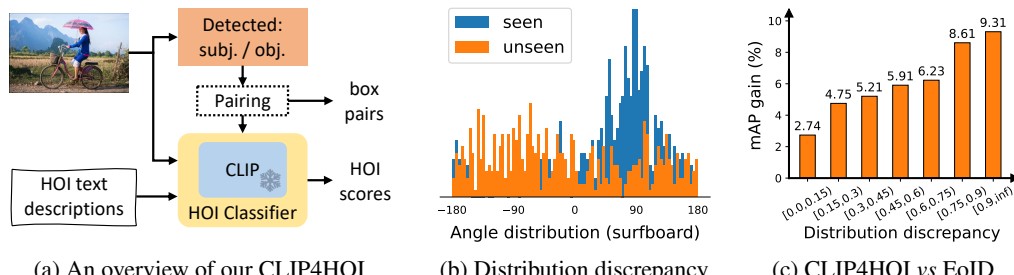

(a) An overview of our CLIP4HOI  (b) Distribution discrepancy  (c) CLIP4HOI *vs* EoID

Figure 1: In our CLIP4HOI, the CLIP model is elaborately adapted for fine-grained pairwise HOI discrimination, avoiding data-sensitive knowledge distillation (a). Thanks to the distribution-agnostic proposal generation, our CLIP4HOI demonstrates better robustness when encountering large positional distribution discrepancy between seen and unseen HOI categories (b)(c). The distribution statistics are the angles between the line from the person to the object and the x-axis. The distribution discrepancy is measured with KL divergence.

First, the prevailing approaches [31, 46] adhere to the one-stage framework, where each query or pair thereof is required to jointly localize the human and the object. This strategy bears the inherent risk of overfitting the decoder to the joint positional distribution of human-object pairs for seen HOI categories, which results in compromised efficacy of the model when encountering novel HOIs that exhibit significant distribution discrepancy with the seen categories, as shown in Figure 1b. Second, the knowledge of CLIP is transferred to the detector through knowledge distillation, the effectiveness of which largely depends on the completeness of the training data. However, given the absence of unseen categories during training, the distillation process is dominated by samples from seen categories, resulting in impaired generalization of the model to unseen categories. Although the latest approach [46] tries to mine potential unseen categories during training, the text descriptions of unseen categories are required in advance. As a result, the generalization capability of the detector is restricted to predefined unseen HOI categories, which is not conducive to practical applications.

To this end, we propose CLIP4HOI, a novel framework for zero-shot generalizable HOI detection. On the one hand, in view of the overfitting issue associated with the joint localization of humans and objects, we seek to tackle the problem of zero-shot HOI detection in a two-stage paradigm, as shown in Figure 1a. Specifically, with the help of an off-the-shelf object detector, the first stage focuses on the accurate identification of humans and objects from images, followed by traversing all feasible human-object combinations in the Human-Object (HO) interactor for pairwise HOI proposal generation. On the other hand, instead of employing data-sensitive knowledge distillation from CLIP to existing detectors, we demonstrate that the adapted CLIP is a natural fit for generalizable fine-grained HOI discrimination. Specifically, based on the generated pairwise proposals, a transformer-based HOI decoder is designed to aggregate contextual visual clues from the output of the CLIP image encoder. HOI scores are then obtained in the HOI classifier by comparing the text embeddings of prompted HOI descriptions with the visual features from different granularities. The final scores for all human-object pairs are the integration of HOI scores and the object class priors out of the object detector. Thanks to the distribution-agnostic pairwise proposal generation and the strong transferability of CLIP, the resulting HOI detector exhibits better robustness to positional distribution discrepancy and superior generalization capability to unseen HOI categories, as shown in Figure 1c.

To verify the effectiveness, we conducted extensive experiments on two prevalent HOI detection benchmarks, *i.e.*, HICO-DET [4] and V-COCO [14]. Results show that our approach exhibits superior performance under a variety of zero-shot settings. Specifically, compared to previous state-of-the-art methods GEN-VLKT [31] and EoID [46], our CLIP4HOI achieves absolute mAP gains of 6.43, 4.67, and 8.41 on the unseen categories of RF-UC, NF-UC, and UO settings, respectively. Moreover, our CLIP4HOI also demonstrates exceptional performance under the fully supervised setting, particularly in rare categories where it outperforms Liu *et al.* [32] by 3.65 mAP on the HICO-DET dataset.

Overall, we make the following three-fold contributions:

- Given the compromised generalization capability of the joint human-object localization, we propose a novel two-stage framework, termed CLIP4HOI, for zero-shot HOI detection, which leverages the generalizable knowledge of CLIP for unseen interaction identification.

- Instead of relying on data-sensitive knowledge distillation, we demonstrate that by carefully designing the adaptation modules, the CLIP model itself exhibits a strong capability for fine-grained HOI discrimination, enabling the unrivaled zero-shot transferability of our CLIP4HOI.
- We conduct extensive experiments on two prevalent benchmarks to verify the effectiveness of the proposed approach. Compared to previous methods, our CLIP4HOI exhibits remarkable performance and sets a series of state-of-the-art records.

## 2 Related Work

**Human-Object Interaction Detection:** There are two mainstream solutions for HOI detection, one-stage and two-stage. Two-stage methods [4, 10–12, 15, 25, 26, 28, 29, 33, 37, 42, 47, 56] first detect human and object instances in the image with an off-the-shelf object detector. Then relation modeling strategies like multi-steam fusion [4] and graph reasoning [10] are designed to identify interactions between each pair of detected humans and objects. One-stage methods, on the other hand, undertake detection, association, and classification in a single stage. Early one-stage methods adopt interaction point [30, 45] or union box [22] as anchors to facilitate the joint localization of humans and objects. Recent one-stage approaches [5, 6, 21, 23, 24, 40, 41, 43, 52, 54, 57] have been inspired by the query-based object detector [3], wherein a group of learnable queries is employed in the transformer decoder to simultaneously predict $\langle human, verb, object \rangle$ triplets. Despite the promising performance under the fully-supervised setting, we argue that such a joint localization strategy tends to overfit the decoder to the joint position distribution of humans and objects for known verb-object combinations. Therefore in this work, we seek to solve the problem of zero-shot HOI detection in a more generalizable two-stage paradigm.

**Contrastive Language-Image Pre-Training:** Contrastive Language-Image Pre-Training (CLIP) [38] is a multimodal learning framework that entails joint training of an image encoder and a text encoder on large-scale image-text pairs. While traditional approaches rely exclusively on either visual or textual cues for learning, CLIP utilizes both modalities to acquire a shared representation space that aligns images and their corresponding captions in a common embedding space. The outcome of this joint training is the ability of CLIP to facilitate zero-shot and few-shot learning on a variety of downstream tasks, including image classification [38], object detection [8, 9, 13], and image retrieval [1]. The success of CLIP also opens up new avenues for zero-shot HOI detection.

**Zero-shot Human-Object Interaction Detection:** Zero-shot human-object interaction (HOI) detection strives to develop a detector that can effectively generalize to HOI categories not encountered during training. It is of practical significance due to the flexibility of verb-object combinations.

Early works [2, 15, 17–20, 34, 36] mainly adopt compositional learning for zero-shot HOI detection. VCL [17] decomposes HOI representation into object- and verb-specific features and then composes novel HOI samples in the feature space via stitching the decomposed features. FCL [19] introduces an object fabricator to generate large-scale HOI samples for rare and unseen categories. ATL [18] decouples HOI representation into a combination of affordance and object representation. Novel interactions are then discovered by combining affordance representation and novel object representation from additional object images. ConsNet [34] proposes a knowledge-aware framework for explicit relation modeling among objects, actions, and interactions in an undirected graph. It leverages graph attention networks to enable knowledge propagation among HOI categories and their constituents. SCL [20] devises a self-compositional learning framework for HOI concept discovery.

Recently, the development of multimodal learning has led to a surge of interest in transferring knowledge from pre-trained vision-language models like CLIP [38] to existing HOI detectors for high-performance zero-shot HOI detection. GEN-VLKT [31] extracts CLIP text embeddings for prompted HOI labels to initialize the classifier and leverages the CLIP visual feature to guide the learning of interactive representation. EoID [46] distills the distribution of action probability from CLIP to the HOI model and detects potential action-agnostic interactive human-object pairs by applying an interactive score module combined with a two-stage bipartite matching algorithm. To avoid data-sensitive knowledge distillation and facilitate better transferability, in our approach, the CLIP model is directly adapted as a fine-grained classifier for generalizable HOI identification.

As a concurrent work, HOICLIP [35] adopts the one-stage design following GEN-VLKT [2] and proposes query-based knowledge retrieval for efficient knowledge transfer from CLIP to HOI detection tasks. During evaluation, it exploits zero-shot CLIP knowledge as a training-free enhancement.

Differently, our CLIP4HOI leverages the two-stage proposal generation strategy to mitigate the overfitting of the method to the joint positional distribution of human-object pairs during training.

## 3 Preliminary

### 3.1 Problem Formulation

In this section, we introduce the problem formulation of zero-shot HOI detection. Let $\mathbb{V} = \{v_1, v_2, \cdots, v_{N_v}\}$ be the set of action verbs and $\mathbb{O} = \{o_1, o_2, \cdots, o_{N_o}\}$ be the set of objects that can be interacted with. The conventional fully-supervised solutions try to incorporate samples of all feasible verb-object pairs $\mathbb{C} = \{(v_i, o_j)|v_i \in \mathbb{V}; o_j \in \mathbb{O}\}$ into the training set and assign each sample a discrete label for to learn a closed-set classifier. Given the inherent flexibility of verb-object combinations, it is impractical to collect a complete HOI dataset. Consequently, researchers have pursued the zero-shot HOI detection paradigm as an alternative direction, where the detector needs to generalize well to unseen categories $\mathbb{C}_{\text{unseen}}$ during the test. Let $\mathbb{V}_{\text{seen}} \subset \mathbb{V}$, $\mathbb{O}_{\text{seen}} \subset \mathbb{O}$, and $\mathbb{C}_{\text{seen}} \subset \mathbb{C} \setminus \mathbb{C}_{\text{unseen}}$ denote seen verbs, seen objects, and seen categories during training, respectively. According to whether verbs and objects in the unseen categories $\mathbb{C}_{\text{unseen}}$ exist during training, zero-shot HOI detection can be divided into three settings: (1) Unseen Composition (UC), where for all $(v_i, o_j) \in \mathbb{C}_{\text{unseen}}$, we have $v_i \in \mathbb{V}_{\text{seen}}$ and $o_j \in \mathbb{O}_{\text{seen}}$; (2) Unseen Object (UO), where for all $(v_i, o_j) \in \mathbb{C}_{\text{unseen}}$, we have $v_i \in \mathbb{V}_{\text{seen}}$ and $o_j \notin \mathbb{O}_{\text{seen}}$; (3) Unseen Verb (UV), where for all $(v_i, o_j) \in \mathbb{C}_{\text{unseen}}$, we have $v_i \notin \mathbb{V}_{\text{seen}}$ and $o_j \in \mathbb{O}_{\text{seen}}$.

### 3.2 Revisiting DETR

DEtection TRansformer (DETR) [3] is an end-to-end object detector that approaches object detection as a direct set prediction problem. At the core of DETR is a transformer encoder-decoder architecture, which is responsible for processing the input image and producing a set of object queries that correspond to potential objects in the image. DETR uses a bipartite matching algorithm that associates the object queries with ground-truth objects during training. Thanks to this query-based detection paradigm, we can easily obtain the features corresponding to each detected instance, enabling efficient extraction of pairwise representation for interaction recognition.

## 4 Methodology

### 4.1 Framework Overview

In this section, we give an overview of the proposed framework. As shown in Figure 2, The proposed CLIP4HOI comprises three key components: Human-Object interactor (HO interactor), HOI decoder, and HOI classifier. Given a single image as input, all humans and objects are first detected using a pre-trained transformer-based detector DETR [3]. Then, the HO interactor takes the decoded query features that correspond to individual human-object pairs as input and derive pairwise HOI tokens through feature interaction and pairwise spatial information injection. After that, with pairwise HOI tokens serving as the input queries, the HOI decoder is engaged to generate pairwise HOI visual features by aggregating the contextual visual clues from the output of the CLIP image encoder. Finally, in the HOI classifier, global and pairwise HOI scores are obtained by comparing the text embeddings of prompted HOI descriptions with the class token from the CLIP image encoder and the decoded pairwise HOI visual features, respectively.

### 4.2 HO Interactor

Following [51], the detections from DETR are first post-processed with non-maximum suppression and thresholding. The filtered result is denoted as $\{\mathbf{B}, \mathbf{X}, \mathbf{C}, \mathbf{S}\}$, where $\mathbf{B} \in \mathbb{R}^{N_d \times 4}$, $\mathbf{X} \in \mathbb{R}^{N_d \times C_d}$, $\mathbf{C} \in \{0, 1, \cdots, N_c - 1\}^{N_d}$, and $\mathbf{S} \in [0, 1]^{N_d}$ denote bounding boxes, decoded query features, object classes, and confidence scores, respectively. $N_d$ denotes the number of detected instances. Then, we perform feature interaction between instances and traverse all valid human-object pairs to generate pairwise HOI tokens $\mathbf{Q}$. These are done through the Human-Object Interactor (HO Interactor). Specifically, the HO Interactor borrows the design of the lightweight interaction head in UPT [51], which consists of a cooperative layer and a competitive layer. Given the detected boxes, unary and

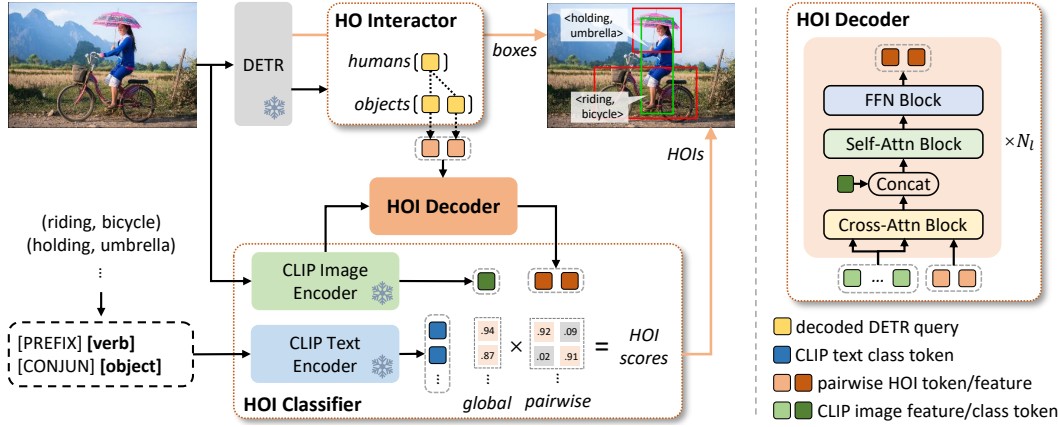

Figure 2: The pipeline of the proposed CLIP4HOI. CLIP4HOI is developed based on the prevalent DETR [3] with three new components, *i.e.*, HO interactor, HOI decoder, and HOI classifier. Based on the detections from DETR, the HO interactor generates pairwise HOI tokens by feature interaction and spatial information injection. The HOI decoder then aggregates the CLIP image representation to produce pairwise HOI visual features, with the pairwise HOI tokens serving input queries. Finally, in the HOI classifier, global and pairwise HOI scores are obtained by comparing the text embeddings of HOI descriptions with the CLIP image class token and the pairwise HOI visual features, respectively.

pairwise spatial features [50] (box center, width, height, pairwise IOU, relative area, and direction) are first extracted and mapped into pairwise positional encodings $\mathbf{E} \in \mathbb{R}^{N_d \times N_d \times C_r}$. Then, decoded query feature $\mathbf{X} \in \mathbb{R}^{N_d \times C_d}$ of detected instances are fed into the cooperative layer along with $\mathbf{E}$ for feature interaction and pairwise spatial information injection:

$$\hat{\mathbf{X}} = \texttt{CoopLayer}(\mathbf{X}, \mathbf{E}) \in \mathbb{R}^{N_d \times C_r}. \tag{1}$$

After that, the competitive layer generates pairwise HOI tokens $\mathbf{Q}$ for all valid human-object pairs:

$$\mathbf{idx}_{\text{valid}} = \{(u, v) | u \neq v, c_u = \text{"human"}\}, \tag{2}$$

$$\mathbf{X}_{\text{pair}} \in \mathbb{R}^{N_{\text{pair}} \times 2C_r}, \quad \text{where } \mathbf{X}_{\text{pair}}^{[i]} = \texttt{CAT}(\hat{\mathbf{X}}^{[u]}, \hat{\mathbf{X}}^{[v]}), \ (u, v) = \mathbf{idx}_{\text{valid}}^{[i]}, \tag{3}$$

$$\mathbf{E}_{\text{pair}} \in \mathbb{R}^{N_{\text{pair}} \times C_r}, \quad \text{where } \mathbf{E}_{\text{pair}}^{[i]} = \mathbf{E}^{[u,v]}, \ (u, v) = \mathbf{idx}_{\text{valid}}^{[i]}, \tag{4}$$

$$\mathbf{Q} = \texttt{CompLayer}(\mathbf{X}_{\text{pair}}, \mathbf{X}_{\text{glob}}, \mathbf{E}_{\text{pair}}) \in \mathbb{R}^{N_{\text{pair}} \times 2C_r}, \tag{5}$$

where $\mathbf{idx}_{\text{valid}}$ is the index set of valid human-object pairs and has size $N_{\text{pair}}$. $\mathbf{X}_{\text{glob}} \in \mathbb{R}^{1 \times C_b}$ is the global visual feature from the backbone of DETR [3]. Please refer to [51] and also the supplementary material for more technical details of the HO interactor.

## 4.3 HOI Decoder

After obtaining the pairwise HOI tokens, we feed them into the HOI decoder to aggregate contextual features from the CLIP image encoder, yielding pairwise HOI visual features. Specifically, the HOI decoder is stacked by $N_l$ customized transformer layers, each of which contains a cross-attention block, a self-attention block, and a feed-forward network (FFN) block. Given the collection of pairwise HOI tokens $\mathbf{Q} \in \mathbb{R}^{N_{\text{pair}} \times 2C_r}$, patch level CLIP image features $\mathbf{F} \in \mathbb{R}^{N_{\text{patch}} \times C_c}$, and the global CLIP image class token $\mathbf{I} \in \mathbb{R}^{1 \times C_c}$, the pairwise HOI visual features $\mathbf{P} \in \mathbb{R}^{N_{\text{pair}} \times C_c}$ are computed as follows:

$$\mathbf{H}_0 = \texttt{Linear}(\mathbf{Q}), \tag{6}$$

$$\mathbf{H}'_l = \texttt{MHCA}(\texttt{LN}(\mathbf{H}_{l-1}), \texttt{LN}(\mathbf{F}), \texttt{LN}(\mathbf{F})) + \mathbf{H}_{l-1}, \qquad l \in 1, \cdots, N_l \tag{7}$$

$$\mathbf{H}''_l = \texttt{MHSA}(\texttt{LN}(\texttt{CAT}(\mathbf{I}, \mathbf{H}'_l))) + \texttt{CAT}(\mathbf{I}, \mathbf{H}'_l), \qquad l \in 1, \cdots, N_l \tag{8}$$

$$\mathbf{H}_l = \texttt{MLP}(\texttt{LN}(\mathbf{H}''^{[1:]}_l)) + \mathbf{H}''^{[1:]}_l, \qquad l \in 1, \cdots, N_l \tag{9}$$

$$\mathbf{P} = \mathbf{H}_{N_l}, \tag{10}$$

where Linear denotes the linear projection $\mathcal{F}_{\text{proj}}(\cdot) : \mathbb{R}^{2C_r} \mapsto \mathbb{R}^{C_c}$ that maps the pairwise HOI tokens into CLIP's joint feature space. LN, MHCA, MHSA, MLP, and CAT denote layer normalization, multi-head cross-attention, multi-head self-attention, multilayer perceptron, and concatenation, respectively. So far, the obtained pairwise HOI visual features $\mathbf{P}$ integrate the relative positional information between humans and objects as well as the contextual visual clues from the large-scale pre-trained CLIP model, greatly promoting the subsequent fine-grained HOI recognition.

## 4.4 HOI Classifier

Given the notable transferability of the CLIP model, this study endeavors to adapt it as a fine-grained classifier for transferable HOI detection. Thanks to the shared embedding space of image and text learned by CLIP after large-scale contrastive pre-training, we can perform fine-grained HOI recognition for both seen and unseen HOI categories by transforming their verb-object combinations into textual descriptions and comparing their embeddings with pairwise HOI visual features. Considering that the manual prompt lacks task-specific heuristics and robustness [55], we choose to train a learnable prompt for text embedding generation. Specifically, we insert several learnable tokens before the verb and object of each HOI category following [44,55], which we denote as [PREFIX] and [CONJUN], respectively. We denote the obtained text embedding as $\mathbf{T} \in \mathbb{R}^{N_c \times C_o}$ ($N_c$ denotes the number of HOI categories). As shown in Figure 2, we compute both global and pairwise HOI scores as follows:

$$\mathbf{I}' = \texttt{Linear}(\mathbf{I}), \quad \mathbf{P}' = \texttt{Linear}(\mathbf{P}), \tag{11}$$

$$\hat{\mathbf{T}} = \texttt{Norm}(\mathbf{T}), \quad \hat{\mathbf{I}} = \texttt{Norm}(\mathbf{I}'), \quad \hat{\mathbf{P}} = \texttt{Norm}(\mathbf{P}'), \tag{12}$$

$$\mathbf{S}_{\text{glob}} = \texttt{Sigmoid}(\hat{\mathbf{I}}\hat{\mathbf{T}}^{\top}/\tau), \quad \mathbf{S}_{\text{pair}} = \texttt{Sigmoid}(\hat{\mathbf{P}}\hat{\mathbf{T}}^{\top}/\tau), \tag{13}$$

where Linear denotes the linear projection head $\mathcal{F}_{\text{proj}}(\cdot) : \mathbb{R}^{C_c} \mapsto \mathbb{R}^{C_o}$, Norm denotes L2 normalization operation, and $\tau$ is the learnable scale factor that rescales the intermediate logits.

The obtained global and pairwise scores $\mathbf{S}_{\text{glob}} \in [0,1]^{1 \times N_c}$ and $\mathbf{S}_{\text{pair}} \in [0,1]^{N_{\text{pair}} \times N_c}$ indicate the presence of HOI categories in the whole image and each human-object pair, respectively. Since each human-object pair may correspond to multiple HOI categories, we adopt the Sigmoid function when computing the HOI scores instead of Softmax with mutually exclusive properties. During inference, the two HOI scores along with the class priors from the object detector are integrated to produce the final classification result for each human-object proposal.

## 4.5 Training and Inference

**Training:** For the global HOI score, the training label is the union of all seen HOI categories present in the image. For the pairwise HOI score, we compare each proposal (detected human-object pair) to the seen HOI targets. When the IOU of both the human and the object exceeds a certain threshold, we consider the proposal a positive sample for the compared target and assign the corresponding label to it. Note that a single proposal may correspond to multiple HOI labels. Considering the scarcity of positive samples, we adopt binary focal loss for training. The final training loss is formulated as the combination of global loss and pairwise loss:

$$\mathcal{L}_{\text{final}} = \texttt{FocalBCE}(\mathbf{S}_{\text{glob}}, \mathbf{Y}_{\text{glob}}) + \beta \cdot \texttt{FocalBCE}(\mathbf{S}_{\text{pair}}, \mathbf{Y}_{\text{pair}}), \tag{14}$$

where $\mathbf{Y}_{\text{glob}} \in \{0,1\}^{1 \times N_c}$ and $\mathbf{Y}_{\text{pair}} \in \{0,1\}^{N_{\text{pair}} \times N_c}$ are global and pairwise labels, respectively. $\beta$ is a hyper-parameter that adjusts the weight of pairwise loss.

**Inference:** Let $\mathbf{S}_{\text{human}} \in [0,1]^{N_{\text{pair}} \times 1}$ and $\mathbf{S}_{\text{object}} \in [0,1]^{N_{\text{pair}} \times 1}$ be the confidence scores out of DETR for paired humans and objects. During inference, the final HOI scores $\mathbf{S}_{\text{final}} \in [0,1]^{N_{\text{pair}} \times N_c}$ are obtained by element-wise multiplication (with broadcasting) of $\mathbf{S}_{\text{glob}}$, $\mathbf{S}_{\text{pair}}$, $\mathbf{S}_{\text{human}}$, and $\mathbf{S}_{\text{object}}$:

$$\mathbf{S}_{\text{final}} = \mathbf{S}_{\text{glob}} \cdot \mathbf{S}_{\text{pair}} \cdot \mathbf{S}_{\text{human}}^{\lambda} \cdot \mathbf{S}_{\text{object}}^{\lambda}, \tag{15}$$

where a hyper-parameter $\lambda$ is introduced to suppress overconfident detections following [50,51]. $N_c$ is equal to the number of seen/full HOI categories during training/test. Based on the object class priors provided by DETR, scores of irrelevant HOIs in $\mathbf{S}_{\text{final}}$ are manually set to 0.

Table 1: Zero-shot HOI detection results on HICO-DET. UC, UO, and UV denote unseen composition, unseen object, and unseen verb settings, respectively. RF- and NF- denote rare first and non-rare first.

(a) UC & UO & UV

| Method | Setting | Full | Seen | Unseen |
|---|---|---|---|---|
| Shen *et al.* [39] | UC | 6.26 | - | 5.62 |
| FG [2] | UC | 12.26 | 12.60 | 10.93 |
| ConsNet [34] | UC | 19.81 | 20.51 | 16.99 |
| EoID [46] | UC | 28.91 | 30.39 | 23.01 |
| HOICLIP [35] | UC | **32.99** | **34.85** | 25.53 |
| **CLIP4HOI** | UC | 32.11 | 33.25 | **27.71** |
| FCL* [19] | UO | 11.43 | 13.71 | 0.00 |
| ATL* [18] | UO | 13.08 | 14.69 | 5.05 |
| FCL [19] | UO | 19.87 | 20.74 | 15.54 |
| ATL [18] | UO | 20.47 | 21.54 | 15.11 |
| GEN-VLKT [31] | UO | 25.63 | 28.92 | 10.51 |
| HOICLIP [35] | UO | **28.53** | **30.99** | 16.20 |
| **CLIP4HOI*** | UO | 28.44 | 30.34 | 18.92 |
| **CLIP4HOI** | UO | 32.58 | 32.73 | 31.79 |
| GEN-VLKT [31] | UV | 28.74 | 30.23 | 20.96 |
| EoID [46] | UV | 29.61 | 30.73 | 22.71 |
| HOICLIP [35] | UV | **31.09** | **32.19** | 24.30 |
| **CLIP4HOI** | UV | 30.42 | 31.14 | **26.02** |

(b) RF-UC & NF-UC

| Method | Setting | Full | Seen | Unseen |
|---|---|---|---|---|
| VCL [17] | RF-UC | 21.43 | 24.28 | 10.06 |
| ATL [18] | RF-UC | 21.57 | 24.67 | 9.18 |
| FCL [19] | RF-UC | 22.01 | 24.23 | 13.16 |
| SCL [20] | RF-UC | 28.08 | 30.39 | 19.07 |
| RLIP [48] | RF-UC | 30.52 | 33.35 | 19.19 |
| GEN-VLKT [31] | RF-UC | 30.56 | 32.91 | 21.36 |
| EoID [46] | RF-UC | 29.52 | 31.39 | 22.04 |
| HOICLIP [35] | RF-UC | 32.99 | 34.85 | 25.53 |
| **CLIP4HOI** | RF-UC | **34.08** | **35.48** | **28.47** |
| VCL [17] | NF-UC | 18.06 | 18.52 | 16.22 |
| ATL [18] | NF-UC | 18.67 | 18.78 | 18.25 |
| FCL [19] | NF-UC | 19.37 | 19.55 | 18.66 |
| SCL [20] | NF-UC | 24.34 | 25.00 | 21.73 |
| RLIP [48] | NF-UC | 26.19 | 27.67 | 20.27 |
| GEN-VLKT [31] | NF-UC | 23.71 | 23.38 | 25.05 |
| EoID [46] | NF-UC | 26.69 | 26.66 | 26.77 |
| HOICLIP [35] | NF-UC | 27.75 | 28.10 | 26.39 |
| **CLIP4HOI** | NF-UC | **28.90** | **28.26** | **31.44** |

# 5 Experiments

## 5.1 Implementation Details

We adopt pre-trained DETR as the object detector, with ResNet-50 [16] serving as the backbone. The HOI classifier is built upon the CLIP model, which takes ViT-B/16 [7] as its visual encoder. During training, the parameters of both the DETR and CLIP models are frozen. The HOI decoder has $N_l = 6$ layers. In each layer, the embedding dimension is 768, the head number of the multi-head attention is 12, and the hidden dimension of the feed-forward network is 3072. The prompt lengths for [PREFIX] and [CONJUN] are 8 and 2, respectively. Following [51], the hyper-parameter $\lambda$ is set to 1 during training and 2.8 during inference. Please refer to the supplementary material for the training details.

## 5.2 Compare with the State-of-the-art Methods

**Zero-Shot HOI Detection:** In Table 1, we evaluate the generalization capability of our approach on the HICO-DET [4] dataset. Performance is reported under three zero-shot settings.

Under the Unseen Composition (UC) setting, CLIP4HOI surpasses most previous top-performing methods. Specifically, compared to EoID [46], our CLIP4HOI exhibits relative mAP improvements of 29.17% (**28.47** *vs* 22.04) and 17.44% (**31.44** *vs* 26.77) on unseen categories under rare first UC and non-rare first UC settings, respectively. Notably, our CLIP4HOI also outperforms the concurrent work HOICLIP [35] by considerable margins in terms of the unseen mAP.

For a fair comparison with previous methods, under the Unseen Object (UO) setting, we provide an additional variant of our approach (marked with * in Table 1), which only uses the bounding boxes of the detections as subsequent inputs. Results show that both versions of our CLIP4HOI exhibit outstanding performance. Specifically, our CLIP4HOI surpasses HOICLIP [35] by a relative mAP improvement of 16.79% (**18.92** *vs* 16.20) on the unseen categories.

Following GEN-VLKT [31], EoID [46], and HOICLIP [35], we report the performance of our approach under the Unseen Verb (UV) setting, where the proposed CLIP4HOI also exhibits promising results. It is worth mentioning that compared to the previous state-of-the-art methods, the advantages is mainly reflected in unseen categories, *e.g.*, relative mAP improvements of 14.58% (**26.02** *vs* 22.71) and 7.08% (**26.02** *vs* 24.30) compared to EoID [46] and HOICLIP [35], respectively. This demonstrates the superior generalization capability of our approach.

Table 2: Fully-supervised HOI detection results on HICO-DET and V-COCO test sets.

| Method | Backbone | HICO-DET | | | | | | V-COCO |
| | | Default | | | Known Object | | | $AP^{S2}_{role}$ |
| | | Full | Rare | Non-rare | Full | Rare | Non-rare | |
|---|---|---|---|---|---|---|---|---|
| IDN[NeurIPS20] [27] | ResNet-50 | 23.36 | 22.47 | 23.63 | 26.43 | 25.01 | 26.85 | 60.3 |
| Zou *et al.*[CVPR21] [57] | ResNet-50 | 23.46 | 16.91 | 25.41 | 26.15 | 19.24 | 28.22 | - |
| HOTR[CVPR21] [23] | ResNet-50 | 25.10 | 17.34 | 27.42 | - | - | - | 64.4 |
| ATL[CVPR21] [18] | ResNet-50 | 28.53 | 21.64 | 30.59 | 31.18 | 24.15 | 33.29 | - |
| AS-Net[CVPR21] [6] | ResNet-50 | 28.87 | 24.25 | 30.25 | 31.74 | 27.07 | 33.14 | - |
| QPIC[CVPR21] [40] | ResNet-50 | 29.07 | 21.85 | 31.23 | 31.68 | 24.14 | 33.93 | 61.0 |
| FCL[CVPR21] [19] | ResNet-50 | 29.12 | 23.67 | 30.75 | 31.31 | 25.62 | 33.02 | - |
| GGNet[CVPR21] [53] | ResNet-50 | 29.17 | 22.13 | 30.84 | 33.50 | 26.67 | 34.89 | - |
| SCG[ICCV21] [50] | ResNet-50 | 31.33 | 24.72 | 33.31 | 34.37 | 27.18 | 36.52 | 60.9 |
| UPT[CVPR22] [51] | ResNet-50 | 31.66 | 25.94 | 33.36 | 35.05 | 29.27 | 36.77 | 64.5 |
| CDN[NeurIPS21] [49] | ResNet-50 | 31.78 | 27.55 | 33.05 | 34.53 | 29.73 | 35.96 | 64.4 |
| Iwin[ECCV22] [41] | ResNet-50 | 32.03 | 27.62 | 34.14 | 35.17 | 28.79 | 35.91 | - |
| Liu *et al.*[CVPR22] [32] | ResNet-50 | 33.51 | 30.30 | 34.46 | 36.28 | 33.16 | 37.21 | 65.2 |
| GEN-VLKT[CVPR22] [31] | ResNet-50 | 33.75 | 29.25 | 35.10 | 36.78 | 32.75 | 37.99 | 64.5 |
| HOICLIP[CVPR23] [35] | ResNet-50 | 34.69 | 31.12 | **35.74** | **37.61** | 34.47 | **38.54** | 64.8 |
| **CLIP4HOI** | ResNet-50 | **35.33** | **33.95** | **35.74** | 37.19 | **35.27** | 37.77 | **66.3** |

Table 3: Robustness comparison between our CLIP4HOI and EoID against distribution discrepancy. Performance is evaluated on the unseen categories of HICO-DET under the NF-UC setting.

| KL divergence | [0,0.3) | [0.3,0.6) | [0.6,0.9) | [0.9,∞) |
|---|---|---|---|---|
| EoID [46] | 26.69 | 27.50 | 26.82 | 18.92 |
| **CLIP4HOI** | **30.97** | **33.06** | **34.15** | **28.23** |
| mAP gain | 4.28 | 5.56 | 7.33 | 9.31 |

**Fully-supervised HOI Detection:** This work focuses on the development of a transferable HOI detector. Nonetheless, we also test the efficacy of our CLIP4HOI for fully-supervised HOI detection, as shown in Table 2. For the HICO-DET [4] dataset, the proposed approach demonstrates remarkable performance, outperforming GEN-VLKT [31] by a margin of 1.11 mAP for full categories. It is worth mentioning that the performance improvement mainly comes from the rare categories, where our CLIP4HOI outperforms GEN-VLKT and [32] by margins of 4.70 mAP and 3.65 mAP, respectively. This indicates that the HOI classifier adapted from CLIP can better handle the long-tailed distribution of HOI categories. For the V-COCO [14] dataset, we ignore HOIs defined with no object labels and report the results in Scenario 2. As shown in Table 2, our CLIP4HOI achieves 66.3 role AP, surpassing GEN-VLKT [31] and [32] by margins of 1.8 mAP and 1.1 mAP, respectively. Compared to the concurrent work HOICLIP [35], our approach also shows advantages on rare categories and performs better under the default setting.

**Robustness to Distribution Discrepancy:** In Table 3, we report the mAP gain of CLIP4HOI over previous top-performing EoID [46]. Performance is evaluated on four unseen subsets of the HICO-DET dataset under the NF-UC setting, each of which exhibits varying degrees of positional distribution discrepancy (measured with KL divergence) from its corresponding seen subset. Results show that our CLIP4HOI exhibits better robustness to positional distribution discrepancy. A detailed definition of the distribution discrepancy is in the supplementary material.

## 5.3 Ablation Study

In this section, we conduct several ablative experiments to verify the effectiveness of our CLIP4HOI. Performance is evaluated on HICO-DET under the RF-UC setting.

**Component Analysis:** As shown in Table 4, we first verified the effectiveness of: (1) *HO interactor*. Without the HO interactor, the overall performance drops by 1.1 mAP, verifying the effectiveness of pairwise feature interaction and spatial information injection. (2) *HOI decoder*. The HOI decoder is designed to aggregate relevant CLIP visual features for better feature alignment in interaction recognition, without which the performance drops by 3.54 mAP and 4.67 mAP for seen and unseen categories, respectively.

Table 4: Component analysis of CLIP4HOI on HICO-DET. The performance is evaluated on HICO-DET under the rare first UC (RF-UC) setting.

| Configuration | Full | Seen | Unseen |
|---|---|---|---|
| **Full version** | **34.08** | 35.48 | **28.47** |
| **Global:** | | | |
| w/o HO interactor | 32.98 | 34.36 | 27.43 |
| w/o HOI decoder | 30.31 | 31.94 | 23.80 |
| **HOI decoder:** | | | |
| w/o CLIP class token | 33.50 | 34.85 | 28.10 |
| w/ Mid-level CLIP features | 32.83 | 34.40 | 26.56 |
| **HOI classifier:** | | | |
| w/o Global HOI score | 33.94 | **36.06** | 25.48 |
| w/o Class priors from DETR | 33.13 | 35.24 | 24.68 |

Table 5: Ablative experiments for hyper-parameters like (a) HOI decoder layer number, (b) text prompt length, and (c) training loss weight. The manual prompt is set to "A photo of a person `[verb]` `[object]`". The performance is evaluated on HICO-DET under the rare first UC (RF-UC) setting.

(a) HOI decoder layer number

| $N_l$ | Full | Seen | Unseen |
|---|---|---|---|
| 2 | 33.45 | 34.80 | 28.08 |
| 4 | 33.65 | 34.96 | 28.41 |
| 6 | **34.08** | **35.48** | **28.47** |
| 8 | 33.66 | 35.32 | 27.01 |
| 10 | 33.94 | 35.30 | 28.46 |

(b) Text prompt length

| `[PREFIX]` | `[CONJUN]` | Full | Seen | Unseen |
|---|---|---|---|---|
| Manual | / | 32.77 | 34.99 | 23.92 |
| 4 | 2 | 33.60 | 35.11 | 27.57 |
| 4 | 4 | 33.37 | 35.15 | 26.23 |
| 8 | 2 | **34.08** | **35.48** | **28.47** |
| 8 | 4 | 33.59 | 34.96 | 28.13 |

(c) Training loss weight

| $\beta$ | Full | Seen | Unseen |
|---|---|---|---|
| 0.2 | 33.46 | 35.06 | 27.04 |
| 0.5 | 33.59 | 35.17 | 27.30 |
| 1.0 | **34.08** | **35.48** | **28.47** |
| 2.0 | 33.82 | 35.18 | 28.37 |
| 5.0 | 33.33 | 35.06 | 26.41 |

Then, we explored the input of the HOI decoder: (1) *CLIP class token*. In the HOI decoder, the CLIP class token $\mathbf{I}$ is incorporated as the input of the self-attention block. Without $\mathbf{I}$, the overall performance decreases slightly by 0.58 mAP, which verifies the effectiveness of the CLIP class token for providing holistic representation of input images. (2) *Mid-level CLIP features*. We also train a variant of CLIP4HOI where each layer of the HOI decoder aggregates the corresponding mid-level features of the CLIP visual encoder layer. Experiments show that aggregating mid-level CLIP visual features results in an overall performance drop of 1.25 mAP.

Finally, we discussed the design of the HOI classifier: (1) *Global HOI score*. While the performance of CLIP4HOI on seen categories exhibits a minor improvement of 0.58 mAP upon removing the global HOI score, the efficacy of the model to identify novel interactions deteriorates significantly by 2.99 mAP. (2) *Class priors from DETR*. The removal of the class priors out of DETR caused a consistent decrease in performance, particularly a drop of 3.79 mAP for unseen categories.

**HOI Decoder Layer Number:** In Table 5a, we experimented with different numbers of HOI decoder layers. Results show that a layer number of 6 exhibits the best overall performance. Adding more layers does not bring consistently better performance, but higher computational overhead.

**Text Prompt Length:** As shown in Table 5b, we explored the impact of different prompt lengths for text embedding generation. Results show that the best performance is observed when the lengths of `[PREFIX]` and `[CONJUN]` are set to 8 and 2, respectively. We also experimented with the manual prompt "A photo of a person `[verb]` `[object]`", whose performance is much lower than the learnable one, especially for the unseen categories (23.92 *vs* **28.47** in mAP).

**Training Loss Weight:** We investigate the impact of varying weights for pairwise loss in Table 5c. Results show that the performance of CLIP4HOI is not substantially influenced by the weight of loss functions. A straightforward summation of global and pairwise losses just works well.

## 5.4 Qualitative Visualization

As shown in Figure 3, we visualize some detection results and their corresponding attention maps in the HOI decoder. We can find that: (1) The HOI decoder can effectively attend to the regions associated with human-object interactions, resulting in accurate aggregation of contextual visual clues from the output of the CLIP image encoder. (2) The obtained HOI detector exhibits remarkable discrimination capability for unseen categories.

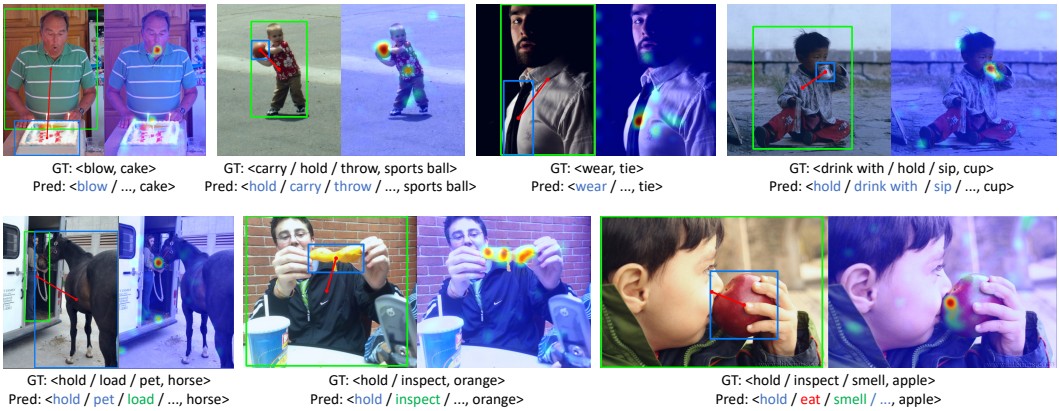

Figure 3: Visualization of detection results and their corresponding attention maps in the HOI decoder. Correctly classified seen and unseen categories are marked in blue and green, respectively. Incorrect recognition results are marked in red. Images are sampled from the HICO-DET dataset. More qualitative results can be found in the supplementary material.

# 6 Conclusion

This paper presents CLIP4HOI, a novel two-stage framework for zero-shot HOI detection, where the generalizable knowledge of the vision-language model CLIP is leveraged for novel interaction identification. By disentangling the detection of humans and objects in the proposal generation phase, the proposed approach exhibits strong robustness to the positional distribution discrepancy between seen and unseen HOI categories. To avoid data-sensitive knowledge distillation and facilitate better knowledge transfer for novel human-object interactions, the CLIP model is elaborately adapted into a fine-grained classifier for pairwise HOI discrimination. Extensive experiments on two prevalent benchmarks verify the effectiveness of the proposed CLIP4HOI framework.

# 7 Acknowledgments

This work was supported by NSFC under Contract U20A20183 and 62021001. It was also supported by GPU cluster built by MCC Lab of Information Science and Technology Institution, USTC, and the Supercomputing Center of the USTC.

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
