# OpenReview forum: "CLIP4HOI: Towards Adapting CLIP for Practical Zero-Shot HOI Detection"
_NeurIPS.cc/2023/Conference — NeurIPS 2023 poster_

### Official Review · Reviewer_Rad1 · 2023-07-04

**Soundness:** 3 good
**Presentation:** 3 good
**Contribution:** 2 fair
**Rating:** 5
**Confidence:** 4

**Summary:**

This paper proposes CLIP4HOI, a two-stage framework for zero-shot HOI detection, where the generalizable knowledge of CLIP is leveraged for interaction classification. To facilitate better transferability and avoid data-sensitive knowledge distillation, CLIP is tuned to a fine-grained HOI classifier. Extensive experiments are conducted on prevalent benchmarks from zero-shot and fully-supervised settings.

**Strengths:**

1.	The motivation of this paper is generally reasonable.
2.	The proposed method is simple and easy to follow.
3.	The experiment results show that the proposed method achieves competitive performance in both fully-supervised and zero-shot settings.


**Weaknesses:**

1.	The novelty of this work is marginal. Some key components are reused in existing HOI detection methods, e.g., HOI interactor is UPT[1]; the learnable text prompt in the HOI classifier is [2].
2.	The paper omits some important baselines, such as HOICLIP[3]. Although HOICLIP is a CVPR2023 paper, it was uploaded to Arxiv in Mar. 2023. The authors should provide a detailed discussion and comparison with it.
3.	The paper misses some experimental results, such as Scenario One on Fully-supervised V-COCO test sets.

[1] Efficient two-stage detection of human-object interactions with a novel unary-pairwise transformer, CVPR 2022.
[2]  Learning transferable human-object interaction detector with natural language supervision, CVPR 2022.
[3]  HOICLIP: Efficient Knowledge Transfer for HOI Detection with Vision-Language Models, CVPR 2023.


**Questions:**

1.	My main concerns and questions lie in the weaknesses. The author should discuss them in detail.
2.	The authors could include more qualitative examples in the supplementary material to illustrate the success and failure cases of their method.


**Limitations:**

The authors addressed the limitations of the proposed method.

---

> ### Author Rebuttal · Authors · 2023-08-08
>
> We are grateful for the insightful comments and clarify the concerns as follows.
>
> ***
>
> Q1: The novelty of this work is marginal. Some key components are reused in existing HOI detection methods.
>
> A1: This work targets to: 1) Alleviate the problem existing in existing top-performing one-stage zero-shot HOI detectors that are prone to overfitting the joint positional distribution of seen human-object pairs during training. 2) Better leverage the general knowledge of the CLIP model for fine-grained HOI discrimination.
> To this end, we adopt a two-stage paradigm and introduce three modules: HO interactor to generate pairwise proposals, HOI decoder to aggregate related features from CLIP image encoder outputs, and HOI classifier for fine-grained HOI discrimination.
> In the implementation of the HO interactor and HOI classifier, we follow some designs of UPT [1] (e.g., interaction module) and [2] (e.g., prompt learning). We would like to clarify that we clearly emphasize their contribution and never take these detailed designs as main innovations of ours.
>
> ***
>
> Q2: The paper omits some important baselines, such as HOICLIP. Although HOICLIP is a CVPR2023 paper, it was uploaded to Arxiv in Mar. 2023. The authors should provide a detailed discussion and comparison with it.
>
> A2: Thanks for pointing out the missed reference. As a common concern with Reviewer baYx, we attempt to address this concern in the global response.
>
> ***
>
> Q3: The paper misses some experimental results, such as Scenario One on Fully-supervised V-COCO test sets.
>
> A3: For fully-supervised testing on the V-COCO dataset, Scenario 1 needs to predict occlusion objects as [0,0,0,0] while Scenario 2 excludes no-object HOI categories during evaluation. This work focuses on the development of a practical zero-shot HOI detector for novel objects, verbs, and object-verb combinations, and thus no special mechanisms are designed to handle occluded objects. Therefore, Scenario 2 better reflects the real fully-supervised performance of the proposed CLIP4HOI on the V-COCO dataset. For completeness, we test the performance of our CLIP4HOI on Scenario 1, and the result is 59.7mAP. This performance is better than that of UPT [1], but lower than that of GEN-VLKT [3], which is reasonable.
>
> ***
>
> Q4: The authors could include more qualitative examples in the supplementary material to illustrate the success and failure cases of their method.
>
> A4: Thanks for your valuable suggestion. We have included more qualitative examples in the one-page pdf of the global response.
>
> ***
>
> References:
>
> [1] Zhang, Frederic Z., Dylan Campbell, and Stephen Gould. "Efficient two-stage detection of human-object interactions with a novel unary-pairwise transformer." Proceedings of the IEEE/CVF Conference on Computer Vision and Pattern Recognition. 2022.
>
> [2] Wang, Suchen, et al. "Learning transferable human-object interaction detector with natural language supervision." Proceedings of the IEEE/CVF Conference on Computer Vision and Pattern Recognition. 2022.
>
> [3] Liao, Yue, et al. "GEN-VLKT: Simplify association and enhance interaction understanding for hoi detection." Proceedings of the IEEE/CVF Conference on Computer Vision and Pattern Recognition. 2022.

---

### Official Review · Reviewer_2e3Y · 2023-07-04

**Soundness:** 2 fair
**Presentation:** 2 fair
**Contribution:** 2 fair
**Rating:** 3
**Confidence:** 5

**Summary:**

This paper proposes a two-stage zero-shot HOI detection paradigm which uses information information from large-scale vision and language models like CLIP. The proposed approach, called CLIP4HOI, first extracts all feasible human-object pairs and generates pairwise proposals. In the second stage, CLIP4HOI uses the CLIP model to classify the pairwise proposals. The authors demonstrate the utility of the proposed approach through experiments on two datasets.

**Strengths:**

Utilizing the information contained in large-scale vision-language models for different tasks is the way to best use such models. The authors have taken inspiration from recent works along these lines and cleverly utilized the CLIP model for HOI proposal classification.

**Weaknesses:**

The motivation for the proposed approach and the differences with prior works are not clear. The paper contains several un-explained terms and claims. The authors should clearly address the questions below in a revision.

**Questions:**

1. The authors claim that prior models over-fit to the join positional distribution of seen human-object pairs. However, I didn't see any convincing evidence to show if this actually happens and if it happens, is it actually a problem. The authors should first show why and how this is a problem. Without a clear demonstration, the motivation behind the proposed approach is not clear.

2. Related to the above, the authors claim that the distribution discrepancy leads to compromised efficacy of the model when encountering novel HOIs. Have the authors actually tested the performance in such cases? It seems figure 1c is trying to do it, but the figure is not clear there's no clear description of the figure (what is the x-axis? what is the class?). The authors should take all classes which demonstrate such distribution dicrepancy and show if the performance is actually compromised.

3. It's not clear how the proposal generation is different from prior works. Prior works also generate all human-object pairs as proposals. How is the proposed approach different from such works?

4. In lines 169-170, the authors talk about "feature interaction". What exactly is feature interaction?

5. Similarly, in several places, the authors talk about "valid" human-object pairs. This is not defined anywhere. What are valid human-object pairs? Are the authors removing any pairs using some heuristics?

6. What is the CoopLayer? What is its form?

**Limitations:**

The authors have not discussed the limitations of their proposed approach. I would recommend that the authors include such a discussion. They can talk about the need to use large-scale vision-language models which require large amount of resources for training. They can also talk about the need for separate object detectors (DETR) which make the overall architecture clunky - there might be ways of reducing the dependence on a large number of additional models.

---

> ### Author Rebuttal · Authors · 2023-08-08
>
> We are grateful for the insightful comments and clarify the concerns as follows.
>
> ***
>
> Q1: The authors claim that prior models over-fit to the join positional distribution of seen human-object pairs. However, I didn't see any convincing evidence to show if this actually happens and if it happens, is it actually a problem. The authors should first show why and how this is a problem. Without a clear demonstration, the motivation behind the proposed approach is not clear.
>
> A1: Thanks for raising this insight concern. We would like to kindly clarify we have discussed this problem in the Section of ``Introduction'' and showed the evidence in Figure 1b. Besides, some more examples are also included in Figure 2 of the supplementary material.
>
> In the literature, previous top-performing methods generally follow the query-based one-stage paradigm, whose query is expected to be used for simultaneously localizing both the human and the object. This kind of prediction paradigm works well under the i.i.d assumption. However, in the zero-shot HOI detection, the joint positional distribution of HOIs is totally different between the training stage and the test stage, making it an o.o.d problem. Therefore, such one-stage methods bear the inherent risk of overfitting to the joint positional distribution of human-object pairs for seen HOI categories during training, which results in compromised efficacy of the model when encountering novel HOIs that exhibit significant distribution discrepancy with the seen categories. We have compared our proposed approach with the previous one-stage method in Figure 1c and Table 2. Results show that our CLIP4HOI exhibits better robustness to positional distribution discrepancy and superior generalization capability to unseen HOI categories.
>
> ***
>
> Q2: Related to the above, the authors claim that the distribution discrepancy leads to compromised efficacy of the model when encountering novel HOIs. Have the authors actually tested the performance in such cases? It seems Figure 1c is trying to do it, but the figure is not clear there's no clear description of the figure (what is the x-axis? what is the class?). The authors should take all classes which demonstrate such distribution discrepancy and show if the performance is actually compromised.
>
> A2: The definition of the x-axis of Figure 1c is the distribution discrepancy between seen and unseen HOI categories. As stated in the caption of Figure. 1, the distribution statistics are the angles (quantized into 90 discrete bins) between the line from the person to the object and the x-axis. The discrepancy is measured with KL divergence. A more detailed explanation can be found in Section 5.2 “Robustness to Distribution Discrepancy” and also in Section A.2 in the supplementary material.
> We indeed take all classes into consideration. Specifically, we first compute the positional distribution discrepancy between the seen and unseen HOI categories corresponding to each object. Then, according to the degree of distribution discrepancy, we divide the whole test set into seven subsets by object. On each subset, we report the performance improvement of our CLIP4HOI compared to the previous method EoID [1], and thus Figure 1c is obtained. Results show that our CLIP4HOI exhibits better performance when encountering larger positional distribution discrepancies (The mAP gain increased from 2.74 to 9.31 as the distribution discrepancy increased).
>
> ***
>
> Q3: It's not clear how the proposal generation is different from prior works. Prior works also generate all human-object pairs as proposals. How is the proposed approach different from such works?
>
> A3: In fact, instead of designing a new HOI proposal generation strategy as the main contribution of this paper, we try to use a traversal-based prior-agnostic HOI proposal generation strategy in a two-stage framework to alleviate the detector from overfitting the joint positional distribution of human-object pairs. As stated in Section 4.2, we borrow the design of the lightweight interaction head in UPT [2] to realize the proposal generation process.
>
> ***
>
> Q4: In lines 169-170, the authors talk about "feature interaction". What exactly is feature interaction?
>
> A4: Feature interaction is done with the HO interactor, which consists of a cooperative layer and a competitive layer. The technical details of these two layers are included in the supplementary material. In a nutshell, the cooperative layer operates on the features of individual human and object instances, while the competitive layer operates on the features of human–object pairs. Both of these two layers are based on the self-attention mechanism (with pairwise positional encoding).
>
> ***
>
> Q5: Similarly, in several places, the authors talk about "valid" human-object pairs. This is not defined anywhere. What are valid human-object pairs? Are the authors removing any pairs using some heuristics?
>
> A5: The definition of valid human-object pairs can be found in Eq. (2)~Eq. (4). We do not heuristically remove any pairs. We just make sure that the subject must belong to the “human” class.
>
> ***
>
> Q6: What is the CoopLayer? What is its form?
>
> A6: As stated in Section 4.2 (line 171~172), the adopted HO Interactor borrows the design of the lightweight interaction head in UPT [2], which consists of a cooperative layer (CoopLayer in Eq. (1)) and a competitive layer (CompLayer in Eq. (5)). For completeness, we have introduced them in detail in Section A.1 of the supplementary material.
>
> ***
>
> References:
>
> [1] Wu, Mingrui, et al. "End-to-end zero-shot hoi detection via vision and language knowledge distillation." Proceedings of the AAAI Conference on Artificial Intelligence. Vol. 37. No. 3. 2023.
>
> [2] Zhang, Frederic Z., Dylan Campbell, and Stephen Gould. "Efficient two-stage detection of human-object interactions with a novel unary-pairwise transformer." Proceedings of the IEEE/CVF Conference on Computer Vision and Pattern Recognition. 2022.

---

### Official Review · Reviewer_baYx · 2023-07-07

**Soundness:** 3 good
**Presentation:** 3 good
**Contribution:** 3 good
**Rating:** 5
**Confidence:** 5

**Summary:**

This paper introduces a new framework to leverage CLIP knowledge for zero-shot HOI detection. The paper proposes an HO interactor for pairwise HOI proposal generation to avoid the overfitting issue associated with the joint localization of humans and objects. Instead of using distillation, this paper directly utilizes CLIP knowledge to overcome the absence of unseen categories during training.

**Strengths:**

* The HOI proposal generation strategy (HO interactor) and the usage of CLIP knowledge (directly using CLIP representation instead of knowledge distillation) are reasonable and effective.
* The overall performance of HOICLIP surpasses its baseline due to its ability to generate novel HOI proposals and efficiently use CLIP information.
* The overall pipeline is clear and easy to understand.

**Weaknesses:**

* The HOI proposal method in this paper (HO interactor) requires traversing all feasible HO combinations for pairwise HOI proposal generation, which may lead to increased computational complexity.
* The motivation behind the specific design of the HOI Decoder is not clear enough.
* A discussion of recent works in CVPR2023 (HOICLIP: Efficient Knowledge Transfer for HOI Detection with Vision-Language Models) is expected.

**Questions:**

* Regarding weakness (1), the author should demonstrate the amount of computational complexity introduced by the process of traversing all feasible HO combinations.

**Limitations:**

* Although the method in this paper is able to avoid the overfitting issue associated with the joint localization of humans and objects, thus improving the ability to detect novel UC, UV, UO HOI pairs, the zero-shot performance may still be limited by the pretrained DETR detector. If the object or human is not detected by the detector, there is no chance that the HOI proposal could be generated by the following modules.

---

> ### Author Rebuttal · Authors · 2023-08-08
>
> We are grateful for the insightful comments and clarify the concerns as follows.
>
> ***
>
> Q1: The HOI proposal method in this paper (HO interactor) requires traversing all feasible HO combinations for pairwise HOI proposal generation, which may lead to increased computational complexity. The author should demonstrate the amount of computational complexity introduced by the process of traversing all feasible HO combinations.
>
> A1: In our approach, the traversal-based proposal generation is adopted to avoid the problem of over-fitting the joint positional distribution of seen human-object pairs during training. We agree that such a traversal-based strategy may bring non-negligible computational overhead. In practice, we follow [1] to filter out detections with scores lower than 0.2, and sample at least 3 and up to 15 humans and objects each. In extreme cases, up to 435 pairwise HOI proposals may be generated. But in most cases, the number of instances (humans + objects) input to the HO interactor is less than 10. Therefore, the computational overhead of our method remains within an acceptable range.
>
> ***
>
> Q2: The motivation behind the specific design of the HOI Decoder is not clear enough.
>
> A2: In this work, the CLIP model is adapted for fine-grained HOI discrimination. To better exploit the joint visual-linguistic space obtained by CLIP pre-training, the visual features of HOI proposals need to be aligned with those of the HOI descriptions, which leads to the introduction of the HOI decoder. In the HOI decoder, given the pairwise HOI tokens produced by the HO interactor, we first perform cross-attention between the tokens and the patch-level CLIP image features to aggregate corresponding CLIP visual features for each HOI proposal. Then, considering that the CLIP image token contains rich global visual cues and is directly aligned with the CLIP text class token during pre-training, we incorporate it in the subsequent self-attention operation for better feature alignment.
>
> ***
>
> Q3: A discussion of recent works in CVPR2023 (HOICLIP: Efficient Knowledge Transfer for HOI Detection with Vision-Language Models) is expected.
>
> A3: Thanks for your valuable suggestion. As a common concern with Reviewer Rad1, we put the answer in the global response letter.
>
> ***
>
> Q4: The zero-shot performance may still be limited by the pre-trained DETR detector. If the object or human is not detected by the detector, there is no chance that the HOI proposal could be generated by the following modules.
>
> A4: Thanks for the insightful concern. We agree it should be a common problem of to-date two-stage zero-shot HOI detectors. We also discuss this limitation of our method in Section B of the supplementary material. Fortunately, given the two-stage design of our CLIP4HOI, a promising solution is to integrate advanced open-vocabulary object detectors like OV-DETR [2] into our CLIP4HOI, as discussed in the supplementary material.
>
> ***
>
> References:
>
> [1] Zhang, Frederic Z., Dylan Campbell, and Stephen Gould. "Efficient two-stage detection of human-object interactions with a novel unary-pairwise transformer." Proceedings of the IEEE/CVF Conference on Computer Vision and Pattern Recognition. 2022.
>
> [2] Zang, Yuhang, et al. "Open-vocabulary DETR with conditional matching." European Conference on Computer Vision. Cham: Springer Nature Switzerland, 2022.

---

### Official Review · Reviewer_dBok · 2023-07-10

**Soundness:** 3 good
**Presentation:** 3 good
**Contribution:** 3 good
**Rating:** 6
**Confidence:** 3

**Summary:**

The paper addresses the problem of Zero-shot Human Object Interaction (HOI) detection, which aims to detect bounding boxes of both seen and unseen interactions.
To achieve this, the paper leverages semantic knowledge from pretrained CLIP models to recognize novel combinations of object-verb as well as unseen verbs or objects.
The main novelty of the work is to propose a two-stage detection framework that processes all pairs of human-object via a Human Object Interactor to avoid overfitting on interactional positions learned from seen HOI.
Moreover, the paper introduces a fine-grained HOI classifier to directly use CLIP for detection without the need for knowledge distillation.
The paper experiments on HOI datasets of HICO-DET and V-COCO to show its effectiveness.

**Strengths:**

+ The proposed idea of using Human-Object Interactor and prompt learning with HOI classifier to directly adapt CLIP model toward the task of human-object interaction is sensible and interesting. This not only helps to significantly simplify the detection pipeline but also improves detection performance.
+ The paper shows strong improvement in performance compared to SOTA on challenging datasets.
+ The paper is self-contained and easy to follow.

**Weaknesses:**

+ The proposed method seems to be mostly effective with novel combinations of verbs and objects, which have been observed in seen interaction during training but becomes less effective for interactions with unseen objects and especially on unseen verbs. Thus, it seems to show that the proposed method is not yet generalizable toward truly novel HOI detection. Can the paper provide justifications for this?
+ Can the paper explain why removing the Global HOI score achieves the best seen HOI performance in Table 4? Removing Global HOI score supposes to reduce seen class overfit, which should have improved unseen instead of seen class performance.
+ The reviewer finds it surprising that prompt learning significantly improves unseen HOI detection, given that the prompt is learned with seen classes and is unaware of unseen classes. Can the paper justify how the proposed method can leverage this prompt for unseen classes?

**Questions:**

Please refer to the weakness section

**Limitations:**

Sufficiently addressed

---

> ### Author Rebuttal · Authors · 2023-08-08
>
> We are grateful for the insightful comments and clarify the concerns as follows.
>
> ***
>
> Q1: Justification for the generalization ability of the proposed method.
>
> A1: We would like to kindly emphasize the advanced performance of our method that demonstrates the generalization ability. In Table 1, we report the zero-shot performance of our CLIP4HOI under three settings, i.e., unseen composition (UC), unseen object (UO), and unseen verb (UV). For each setting, we report full mAP (including seen HOI and unseen HOI), seen mAP (only including seen HOI), and unseen mAP (only including unseen HOI). It is unseen mAP that reflects the generalization ability of a method towards unseen HOI categories. We would like to clarify our CLIP4HOI achieves the best performance among the compared methods in terms of unseen mAP under all three settings (18.92 mAP vs. 10.51 mAP under the UO setting, 26.02 mAP vs. 22.71 mAP under the UV setting, and 27.71 mAP vs. 23.01 mAP under the UC setting).
>
> ***
>
> Q2: Can the paper explain why removing the Global HOI score achieves the best seen HOI performance in Table 4? Removing Global HOI score supposes to reduce seen class overfit, which should have improved unseen instead of seen class performance.
>
> A2: The global HOI score is responsible for perceiving the HOIs existing in the entire receptive field, which provides discriminative knowledge from a global perspective. The pairwise HOI score only focuses on the perception of human-object interactions corresponding to a provided proposal, which is easily affected by the quality of the proposal generation process. In practice, we found that the pairwise HOI score may sometimes have a high response to the HOI categories that do not exist on the image. Such values are usually lower than the score of the seen HOI categories. Therefore, it hardly affects the discrimination of seen HOI categories. However, for the unseen HOI categories that typically exhibit lower HOI scores than the seen HOI categories, the impact of these noises is relatively large. The global HOI score helps mitigate this interference. Therefore, we argue that the degradations in discrimination capability for unseen categories are justifiable after removing the global HOI score.
>
> ***
>
> Q3: The reviewer finds it surprising that prompt learning significantly improves unseen HOI detection, given that the prompt is learned with seen classes and is unaware of unseen classes. Can the paper justify how the proposed method can leverage this prompt for unseen classes?
>
> A3: The function of prompt learning is to tune the input of the CLIP text encoder into a sentence pattern suitable for a certain downstream task without finetuning the weights of the whole model, which may cause catastrophic forgetting. The learnable prompt we adopt is not limited to seen HOI categories (category-agnostic). Although only seen verbs and objects are available during training, thanks to the general knowledge owned by the large-scale pre-trained CLIP model, the learned prompt can also be used for unseen HOI categories. Moreover, the primary manual prompt such as "a photo of a person xxx" which is commonly used in image recognition was found to be not the optimal solution for the HOI task [3]. Therefore, compared to the manual prompt, the learnable prompt used in our approach leads to significant performance improvements.
>
> ***
>
> References:
>
> [1] Zhou, Kaiyang, et al. "Learning to prompt for vision-language models." International Journal of Computer Vision 130.9 (2022): 2337-2348.
>
> [2] Zhou, Kaiyang, et al. "Conditional prompt learning for vision-language models." Proceedings of the IEEE/CVF Conference on Computer Vision and Pattern Recognition. 2022.
>
> [3] Wang, Suchen, et al. "Learning transferable human-object interaction detector with natural language supervision." Proceedings of the IEEE/CVF Conference on Computer Vision and Pattern Recognition. 2022.

---

> > ### Comment · Reviewer_dBok · 2023-08-21
> >
> > Dear authors,
> >
> > Thanks for clarifying my concerns in the rebuttal.
> > As such, I will keep my original score.

---

> > > ### Author Response · Authors · 2023-08-22
> > > **Thanks for your support**
> > >
> > > Thanks for your valuable support!

---

### Author Rebuttal · Authors · 2023-08-09

The following is a common question raised by Reviewer baYx and Rad1.

***

Q1: The paper omits some important baselines, such as HOICLIP. Although HOICLIP is a CVPR2023 paper, it was uploaded to Arxiv in Mar. 2023. The authors should provide a detailed discussion and comparison with it.

A1: Thanks for pointing out this concurrent work. HOICLIP [1] adopts the one-stage design following GEN-VLKT [2] and proposes query-based knowledge retrieval for efficient knowledge transfer from CLIP to HOI detection tasks. In addition, it exploits zero-shot CLIP knowledge as a training-free enhancement during evaluation. Differently, our CLIP4HOI leverages the two-stage proposal generation strategy to mitigate the overfitting of the method to the joint positional distribution of human-object pairs during training. As for the similarities, HOICLIP and our CLIP4HOI both retain the image encoder of CLIP to better exploit the general knowledge learned by large-scale pre-training, although the implementation details differ.
We compare the performance of our CLIP4HOI with HOICLIP under five zero-shot settings in the following table. Results show that: 1) Under UC, UO, and UV settings, our CLIP4HOI performs on par or slightly inferior in terms of seen mAP. 2) Under all five zero-shot settings, our CIP4HOI outperforms HOICLIP in terms of unseen mAP. This demonstrates that our proposed CLIP4HOI exhibits a stronger generalization ability for unseen HOI categories than HOICLIP. We will follow your suggestion to incorporate this discussion and comparison into the revised manuscript.

|  Method   | Setting  | Full | Seen | Unseen |
|  :----  | :----: |  :----:  | :----: |  :----:  |
| HOICLIP  | UC | **32.99** | **34.85** | 25.53 |
| **CLIP4HOI (Ours)**  | UC | 32.11 | 33.25 | **27.71** |
|   |  |  |  |  |
| HOICLIP  | UO | **28.53** | **30.99** | 16.20 |
| **CLIP4HOI (Ours)**  | UO | 28.44 | 30.34 | **18.92** |
|   |  |  |  |  |
| HOICLIP  | UV | **31.09** | **32.19** | 24.30 |
| **CLIP4HOI (Ours)**   | UV | 30.42 | 31.14 | **26.02** |
|   |  |  |  |  |
| HOICLIP  | RF-UC | 32.99 | 32.99 | 25.53 |
| **CLIP4HOI (Ours)**   | RF-UC | **34.08** | **35.48** | **28.47** |
|   |  |  |  |  |
| HOICLIP  | NF-UC | 27.75 | 28.10 | 26.39 |
| **CLIP4HOI (Ours)**  | NF-UC | **28.90** | **28.26** | **31.44** |

***

References:

[1] Ning, Shan, et al. "HOICLIP: Efficient Knowledge Transfer for HOI Detection with Vision-Language Models." Proceedings of the IEEE/CVF Conference on Computer Vision and Pattern Recognition. 2023.

[2] Liao, Yue, et al. "GEN-VLKT: Simplify association and enhance interaction understanding for hoi detection." Proceedings of the IEEE/CVF Conference on Computer Vision and Pattern Recognition. 2022.

---

### Decision · Program_Chairs · 2023-09-21

**Decision:**

Accept (poster)

**Comment:**

This submission proposes a method for Human-Object Interactor detection. The submission received one weak accept, two borderline accepts and one reject (2e3Y). After rebuttal, 2e3Y didn’t change his / her rating, but clearly mentioned ‘ If the paper is accepted, I would recommend that the authors provide more conclusive evidence for this being a problem.’ Reviewer 2e3Y mentioned that some concerns remain including motivation of the paper is not clear. The AC carefully read the submission, comments, and rebuttals, the AC agree with the majority of the reviewers. As for the last concern from reviewer 2e3Y of motivation, it’s clear this is an interesting topic as reviewer dBok mentioned ‘the proposed idea of using Human-Object Interactor and prompt learning with HOI classifier to directly adapt CLIP model toward the task of human-object interaction is sensible and interesting. This not only helps to significantly simplify the detection pipeline but also improves detection performance.’ Therefore, the AC suggests accepting this submission.